# Assessment of Equine Influenza Virus Status in the Republic of Korea from 2020 to 2022

**DOI:** 10.3390/v15102135

**Published:** 2023-10-23

**Authors:** Seong-In Lim, Min Ji Kim, Min-Ji Kim, Sang-Kyu Lee, Hyoung-Seok Yang, MiJung Kwon, Eui Hyeon Lim, In-Ohk Ouh, Eun-Jung Kim, Bang-Hun Hyun, Yoon-Hee Lee

**Affiliations:** 1Viral Disease Division, Animal and Plant Quarantine Agency, Gimcheon 39660, Republic of Korea; saint78@korea.kr (S.-I.L.); mingming0525@korea.kr (M.J.K.); mj852@hanmail.net (M.-J.K.); alwjd3920@naver.com (M.K.); sksms147zld@korea.kr (E.H.L.); kejpo30@korea.kr (E.-J.K.); hyunbh@korea.kr (B.-H.H.); 2Veterinary Center, Korea Racing Authority, Gwacheon 13822, Republic of Korea; bestvet@kra.co.kr; 3Animal Health Diagnosis Division, Jeju Self-Governing Provincial Veterinary Research Institute, Jeju 63344, Republic of Korea; hsypath@korea.kr; 4Division of Vaccine Development Coordination, National Institute of Infectious Disease, Cheongju 28160, Republic of Korea; dvmoio@korea.kr

**Keywords:** surveillance, equine, influenza A virus, H3N8, vaccination

## Abstract

Equine influenza virus (EIV) causes acute respiratory disease in horses and belongs to the influenza A virus family *Orthomyxoviridae*, genus *Orthomyxovirus*. This virus may have severe financial implications for the horse industry owing to its highly contagious nature and rapid transmission. In the Republic of Korea, vaccination against EIV has been practiced with the active involvement of the Korea Racing Authority since 1974. In this study, we monitored the viral RNA for EIV using PCR, as well as the antibody levels against ‘A/equine/South Africa/4/03 (H3N8, clade 1)’, from 2020 to 2022. EIV was not detected using RT-PCR. The seropositivity rates detected using a hemagglutination inhibition assay were 90.3% in 2020, 96.7% in 2021, and 91.8% in 2022. The geometric mean of antibody titer (GMT) was 83.4 in 2020, 135.7 in 2021, and 95.6 in 2022. Yearlings and two-year-olds in training exhibited lower positive rates (59.1% in 2020, 38.9% in 2021, and 44.1% in 2022) than the average. These younger horses may require more attention for vaccination and vaccine responses against EIV. Continuous surveillance of EIV should be performed to monitor the prevalence and spread of this disease.

## 1. Introduction

Equine influenza virus (EIV) is a major cause of respiratory diseases in horses. EIV is classified within the family *Orthomyxoviridae*, genus *Orthomyxovirus*, influenza type A. EIV is an enveloped virus comprising eight genomic segments of single-strand RNA, designated as a subtype, and determined by the presence of the surface proteins haemagglutinin (HA) and neuraminidase (NA) [1,2]. Owing to the highly contagious nature and rapid spread of EIV, this virus has severe financial implications for the horse industry. The global transportation of horses has been responsible for numerous outbreaks of EIV through the introduction of the virus into previously unexposed horse populations [3,4]. 

Only two EIV subtypes have caused widespread outbreaks: A/equine/1 (H7N7) and A/equine/2 (H3N8) viruses. The H7N7 subtype was first isolated from a horse in Czechoslovakia in 1956 [5]. The H3N8 subtype was first isolated in Miami in 1963. The virus has caused disease outbreaks in Europe, North America, India, and China. While the H7N7 subtype has not been detected in horses after 1979, the H3N8 subtype still causes major economic impacts on the equine industry in most parts of the world [6,7,8].

The H3N8 EIV evolved into two distinct lineages, ‘American’ and ‘European’, in the late 1980s. The American lineage further evolved into the South American, Kentucky, and Florida lineages. The Florida lineages further diverged into two sublineages (Florida clade 1 and clade 2), which have been predominantly circulating worldwide. Florida clade 1 viruses have caused major outbreaks in Africa, Asia, Australia, and Europe, and clade 2 viruses have also spread to Europe and Asia [7,9,10]. 

Vaccination is a key control measure against EI. Whole-inactivated and subunit EI vaccines were the first to be developed, and these have been the most widely used vaccines in recent decades [11]. Based on the antigenic characterization of circulating viruses, EI vaccine strains are recommended by the World Organization for Animal Health (WOAH). The current recommendation of the Expert Surveillance Panel for equine influenza is that vaccines should contain ‘A/equine/South Africa/4/03 (for Florida clade 1)’ and ‘A/equine/Richmond/1/2007 (for Florida clade 2)’ [6,12]. 

Globally, EIV has repeatedly demonstrated that antigenic drift affects vaccine efficacy [13]. Therefore, EI surveillance maintains awareness of the emergence and international spread of antigenic variants. It serves as an early warning system and is fundamental to influenza control programs based on vaccination.

Since 1974, in the Republic of Korea (ROK), EI vaccination has been practiced with the active involvement of the KRA. EI vaccines have been imported and used since 2008. The vaccine uses recombinant canarypoxvirus, which contains a segment of the EIV gene (HA) expressed after injection [9,11]. It contains EIV strain ‘A/equine/Ohio/03 (clade 1)’ and ‘A/equine/Richmond/1/07 (clade 2)’. According to the vaccination regime, horses in the ROK are vaccinated once or twice yearly [9]. In naïve horses, revaccination occurs 4–6 weeks after the primary vaccination. In previously vaccinated horses, revaccination was administered once yearly. In the ROK, EI has been monitored for antibody responses after vaccination [9,14,15,16].

In the ROK, in 2011, H3N8 ‘A/Equine/Kyonggi/SA1/2011 (KG11)’ was isolated from horses showing typical symptoms of respiratory disease, which had a naturally truncated NS1 protein-coding gene. The H3N8 isolate belongs to Florida sublineage clade 1 of the American lineage [17]. However, serological evidence of EIV infection in unvaccinated horses was reported by the Korean National Veterinary Research and Quarantine Service (formerly APQA) in 2010 [14]. Since 2011, EIV has not occurred in the ROK. 

In the ROK, the equine population has gradually increased since the 1980s, and the number of horses raised was 26,525 in 2020, 26,868 in 2021, and 27,631 in 2022. Among horses in the ROK, riding horses for leisure accounted for the largest proportion at 41.4–47.6%, followed by race horses at 28.4–31.4% and breeding horses at 15.5–16.6% from 2020 to 2022 [18]. The number of horses slaughtered was 1143 in 2020, 1270 in 2021, and 1501 in 2022. Meat is used as food or animal feed [19]. Approximately 55.0% of the total number of horses are raised on Jeju Island. Gyeonggi is the second-largest province that raises horses [18]. Therefore, in Gyeonggi and Jeju Island, the number of horses tested was higher than that in other provinces. In addition, two major Korea Racing Authority (KRA) race parks are located on Jeju Island and in Gyeonggi Province (Figure 1). 

In this study, viral RNA was monitored to check the presence of EIV in approximately 2.5–3% of the total number of horses (nasal swabs and lungs), and EI antibodies against H3N8 were monitored in approximately 5% of the total number of horses in the ROK to check EI antibody levels from 2020 to 2022. This study discusses the current status of the ROK after EI vaccination. 

## 2. Materials and Methods

### 2.1. Sample Collection and Preparation

We conducted a surveillance of several horse populations in the ROK. Blood and nasal swabs were obtained from the KRA (Gwacheon, Republic of Korea) and private farms between 2020 and 2022 (Table 1). Blood samples were collected via venipuncture from the jugular vein of horses using a BD vacutainer K2E (EDTA) (BD Biosciences, Franklin Lakes, NJ, USA) centrifuge to collect serum, which was stored at −20 °C until testing. The sera were heat-inactivated in a water bath at 56 °C for 30 min prior to testing. Nasal swabs were collected using NS—1 nasal swab (Noble Biosciences, Gyeonggi, Republic of Korea). We performed centrifugation to collect the supernatant and stored it at −80 °C until further analysis. 

Horse lung tissues were collected from a slaughterhouse on Jeju Island, ROK, from 2020 to 2022. The tissues were stored at −80 °C until testing.

### 2.2. RNA Extraction and RT-PCR

Viral nucleic acids from nasal swabs and lung tissues were prepared using RNeasy Mini Kit (Qiagen, Hilden, Germany), following the manufacturer’s instructions. 

The primers used in this study were designed to align with most type A influenza viruses originating from multiple species. The conventional RT-PCR targets the common sequence of the M gene, according to Lee et al. [20]. 

### 2.3. Hemagglutination Inhibition (HI) Assay

EIV A/equine/South Africa/4/03 (H3N8) (American lineage, Florida sublineage clade 1) was used as an antigen in the HI assay. The virus was kindly provided by Dr. Debra Elton from the former World Organization for Animal Health (WOAH) Reference Laboratory of Equine Influenza, Center for Preventive Medicine, Animal Health Trust, UK [9]. This virus was propagated in 10-day-old embryonated specific-pathogen-free (SPF) eggs (VALO, DD, USA) and incubated at 37 °C for 3 days. The allantoic fluid was harvested after chilling at 4 °C and stored at −70 °C before use.

Virus titers were measured using a hemagglutination (HA) assay as previously described [16]. Briefly, 25 μL of allantoic fluid was serially diluted 2-fold with 25 μL of phosphate-buffered saline (PBS). Fifty microliters of 0.5% chicken red blood cells (RBCs) was added to each well. The virus and RBC mixture were incubated at room temperature (RT) until a distinct RBC button was formed (30–60 min) in the control well.

Antibody titers were measured using the HI assay as previously described [1]. Horse serum samples were treated with 100 μL of 0.016 M potassium periodate at room temperature (RT) for 15 min, and then 50 μL of 3% glycerol in PBS was added. The mixtures were placed at RT for 15 min and then incubated at 56 °C for 30 min. Two-fold diluted serum (25 μL) and the same volume of 4 HA unit antigen were mixed in a 96-well microplate with a V-shaped bottom and incubated at RT for 1 h. Then, 25 μL of 0.5% chicken RBCs was added to each well and incubated at RT for 1 h. The HI titer was determined as the reciprocal of the endpoint dilution that showed complete HI. A HI titer ≥ 10 was used as the cut-off value for seropositivity in all samples [10].

### 2.4. Data Analysis

For the quantitative evaluation of positive horse sera, we calculated the geometric mean of antibody titer (GMT) by averaging the logarithms of the horse titers in the positive sera and then converting the mean into a real number. GMTs were calculated using Microsoft Excel 2016 version 16.0.10402.20023 (Microsoft Corporation, Seattle, WA, USA). The positive rates of horses by region, type, and age were calculated as the number of positive horses divided by the total number of tested horses.

## 3. Results

### 3.1. PCR Detection of Equine Influenza Virus

To test for the presence of viral RNA against EIV in the collected nasal swabs and lung tissues from 2020 to 2022, we examined the samples using conserved influenza A-M gene primers. All nasal swabs and lung tissues tested negative for EIV RNA using RT-PCR.

### 3.2. Detection of EIV Antibody

#### 3.2.1. Seropositivity Rates and Mean Antibody Titers According to Year

The HI assays showed that the tested horses had 92.9% of the antibodies against EIV (H3N8, A/equine/South Africa/4/03) between 2020 and 2022 (Table 2). In 2020, the positive serum count was 90.3% (GMT 83.4) from 1323 sera in total to 1195, 96.7% (GMT 135.7) from 1312 sera to 1269 sera in 2021, and 91.8% (GMT 95.6) from 1346 sera to 1235 sera in 2022. 

#### 3.2.2. Seropositivity Rates and Mean Antibody Titers According to Horse Type

As demonstrated in Table 3 and Figure 2, race horses showed high antibody positivity rates (approximately > 94.0%) in 2020 and 2021, but not in 2022 (82.3%). Riding horses also showed high antibody positivity rates (approximately > 90%) from 2020 to 2022. Broodmares and stallion studs showed higher antibody positivity rates than did other horse types. The positivity rates for broodmares were 93.5% in 2020, 98.6% in 2021, and 93.7% in 2022. Furthermore, the positivity rates in stallion studs were 100% in 2020, 100% in 2021, and 100% in 2022. The training horses were young (under two years of age). They showed lower positivity rates (38.9–59.1%) from 2020 to 2022. Except for training horses, all groups showed relatively high positivity rates (approximately 82.3–100%) from 2020 to 2022. In 2020, the GMTs were 63.6 and 64.0 in broodmares and stallion studs, respectively. In 2021, training horses showed the lowest GMT (78.0); by 2022, race and training horses showed lower GMTs than did the other horse types (Figure 3). 

#### 3.2.3. Seropositivity Rates and Mean Antibody Titers According to Horse Age

The ages of the horses tested varied (from 0 to 26 years), and the results were analyzed after the horses were divided into six groups (0–1, 2, 3–5, 6–10, >10 years old, and unknown) (Table 4 and Figure 4). The youngest (0–1) horses showed the lowest seropositivity rates (29.2–50.0%). The two-year-old horses showed a seropositivity rate of 73.2–75.0%; however, the 2-year-olds showed a much higher positivity rate than the 0–1-year-olds. The 3-year-olds or older showed a higher seropositivity rate (approximately 92.0–100%) except in 2022 (85.6%). Similarly, the 2-year-olds showed the lowest antibody titers (GMT 56.6–110.8) compared to those of other age groups. In 2021, horses of all ages showed the highest antibody titers (GMT 110.8–159.7) (Figure 5). 

#### 3.2.4. Seropositivity Rates and Mean Antibody Titers According to Province

The ROK comprises nine provinces. Approximately 70% of domestic horses are raised on Jeju Island and Gyeonggi Province. As a result, 70% of tested samples were collected on Jeju Island and Gyeonggi Province (Table 5). 

In 2020, Chungnam Province had the lowest seropositivity rate (63.6%) and antibody titer (GMT 31) compared to those of other provinces from 2020 to 2022. According to the provinces in ROK, the seropositivity rates were 63.6–96.6%, and the GMTs were 31–102.2 in 2020. In 2021, the seropositivity rates were 93.9–100%, and the GMTs were 111.8–227.3. In 2022, the seropositivity rates were 89.9–100%, and the GMTs were 71.4–176.7. 

## 4. Discussion

The equine population is highly mobile, and horses travel long distances by road and air for competition and breeding purposes. When an infected horse is introduced into a susceptible population, the spread of the virus can be explosive. This was illustrated in Ireland in 1989 and Australia in 2007. In Ireland, several shows were held with EI-infected horses from several countries, and some horses returned home and to mixed yards, after which the virus spread rapidly to the thoroughbred population. In Australia, the virus initially spread in the general horse population and then spread to the thoroughbred population, and it was estimated that over 75,000 horses were infected despite strict preventive and control measures. After considerable efforts, the disease was eradicated at a significant cost of approximately one billion Australian dollars [13,21,22]. 

Only New Zealand and Iceland, with their significant horse populations, remained free from EIV. Some countries, including Australia and South Africa, have eradicated EIV after past outbreaks. However, EIV is generally considered enzootic in Europe, North and South America, and Asia [7,10,13,22]. EIV outbreaks are reported yearly worldwide and are caused by strains belonging to Florida sublineage clades 1 and 2 [23,24]. EIV occurred in donkeys in China in 2017, and an EIV outbreak in West Africa in 2009 was reported to have caused 60,000 deaths in a highly susceptible population of donkeys [25,26]. EIV infection is rare in donkeys, although they are more sensitive than are other horses [3]. 

Vaccination is the most useful prophylactic strategy; however, the continuous genetic evolution of the virus demands a genetic characterization of currently circulating EIVs for the selection of a candidate vaccine strain. Recurrent vaccination failures against this virus due to antigenic drift and shifts have been disappointing. Because vaccine failures occur in several parts of the world, better vaccines are required to completely eradicate EIV [6,7,11,21]. EIV outbreaks in vaccinated horses have been reported in countries such as Italy, Croatia, and Japan [27,28,29]. In Ireland, in 2014, although there was a clear vaccination history, EIV outbreaks were reported because of the lack of updated vaccines for Florida sublineage clade 2 [30]. In Brazil, in 2015, an EIV outbreak was reported in both vaccinated and unvaccinated horses because of the use of old vaccines without updates or that of updated vaccines without proper trials [31]. 

Vaccine mismatch reduces protection against infection and virus shedding. High antibody levels are required in the field to protect horses from heterologous strains. Epidemics are more likely to occur when vaccines have not been updated [13,32]. Amino acid changes in the antigenic sites between the HA1 subunit of the outbreak strain and the strains used in the vaccines likely accounted for vaccine failure and the same clinical signs in vaccinated and unvaccinated horses [27,28]. To address this problem, continuous checks and monitoring through surveillance programs and updating vaccines with recent strains remain the best and most effective ways to prevent and control this disease.

This study did not detect viral RNA against EIV using RT-PCR between 2020 and 2022. This may mean that wide-type EIV (no vaccine strain) was not in the ROK and that the risk of introduction of EIV was very low. The seropositivity rates against A/equine/South Africa/4/03 (clade 1) were 90.3% in 2020, 96.7% in 2021, and 91.8% in 2022, as found using HI assays. Herds with 75% vaccination coverage exhibited better disease control when exposed to virulent infections [21]. The WOAH-recommended EI-strain-containing vaccine (ProteqFlu, Boehringer Ingelheim Vetmedica GmbH, Binger, Germany) has been used for EI vaccination in the ROK, and this may explain the absence of EI in the ROK. Overall, the EI vaccine is being effectively administered, and EI is well controlled. The GMT against total positive antibodies was 83.4 in 2020, 135.7 in 2021, and 95.6 in 2022. A titer of 1:64 is conventionally viewed as clinically protective in horses; however, this has not been rigorously established, especially against challenges with heterologous strains [7]. Thus# was sufficient for defense against EIVs. According to horse type, training horses showed low positivity rates (38.9–59.1%). The training horses were yearlings (under two years of age). Training horses account for less than 1% of all horses and may be less managed than race, riding, or breeding horses. Since training horses can act as a dangerous section when EIV enters from the outside, more attention should focus on vaccination. According to the age of the horse, the youngest (0–1-year-old) horses showed the lowest seropositivity rates (29.2–50.0%). However, the antibody levels of zero-to-one-year-old horses were greater than GMT 95. In foals, maternal antibodies for influenza generally decrease to a low but detectable level by six months [6]. In the ROK, since vaccination in horses begins at the end of more than a year, the antibody positivity rate is low, and the detected antibody is assumed to be a maternal antibody.

EI causes high morbidity and mortality in foals, horses in poor health, and donkeys. In the ROK, approximately 2% of donkeys are raised as domestic animals [18]. They are primarily used for riding and enjoyment. Although the number of donkeys is small, they can act as a risk zone because they are less managed than are other horses. 

This study was conducted to monitor the status of EIV and antibody levels against the EI vaccine because EI vaccination has been administered twice a year regularly in the ROK, with the active involvement of the KRA. Riding, racing, and breeding horses are generally over 3 years old, and they showed high seropositivity rates and antibody titers. In the ROK, the number of 2-year-olds or younger was 25.8% of the total number of horses in 2022. Recently, EIV has not been detected in the ROK, but 2-year-old or younger horses could be a target of EIV at any time.

Outbreaks of Florida clade 1 in Asia were found to have been introduced t the Americas, as reported from Argentina to Dubai, North America to Japan, and North America to Malaysia. These findings reflect the effect of global horse transport on the spread of H3N8 strains [33]. The risk of introducing EIV is increasing, particularly with the movement of racing and breeding horses. As of 2022, the ROK has imported the largest number of racehorses and breeding horses from the USA [34]. Recently, there have been no EIV outbreaks in the ROK; however, in the USA, EIV strains of the Florida clade 1 subtype have been circulating. Increased outbreaks of EIV were reported in the USA, where the disease is endemic [7,35]. China, Japan, and Mongolia are geographically closest to the ROK. The risk of EIV outbreak in the ROK is low because horses have not recently been imported from these countries. However, continuous monitoring and vaccinations should be performed in the ROK. 

EI surveillance is fundamental for influenza control programs based on vaccination. EI surveillance reduces the economic impact of the disease by maintaining awareness of the emergence and international spread of antigenic variants [13,36]. EIV has resulted in costly damage to the horse industry and can cross the host species barrier from horses to dogs. EIV has low to negligible zoonotic potential. However, in experimental settings, EIV has shown the ability to infect humans, and few people in contact with infected horses have developed antibodies against EIV. The isolation of equine H3N8 from humans has not yet been confirmed [11,37]. 

Regular monitoring and surveillance of EIV can also help prevent and protect equine populations from EIV, considering the evolution of the virus [36]. The increased international movement of horses for breeding and competition is an important factor in the global spread of EI worldwide. Therefore, in the ROK, continuous surveillance of EIV should be practiced to monitor the introduction of this disease, and an EIV vaccine should be developed for specific strains.

## 5. Conclusions

This study, the purpose of which is to evaluate the status of EIV using antibody levels against the EI vaccine, confirms that EI vaccination shows high seropositivity rates and antibody titers. Additionally, wild-type EIV was not detected using RT-PCR. Therefore, continuous vaccination against EIV should be practiced.

## Figures and Tables

**Figure 1 viruses-15-02135-f001:**
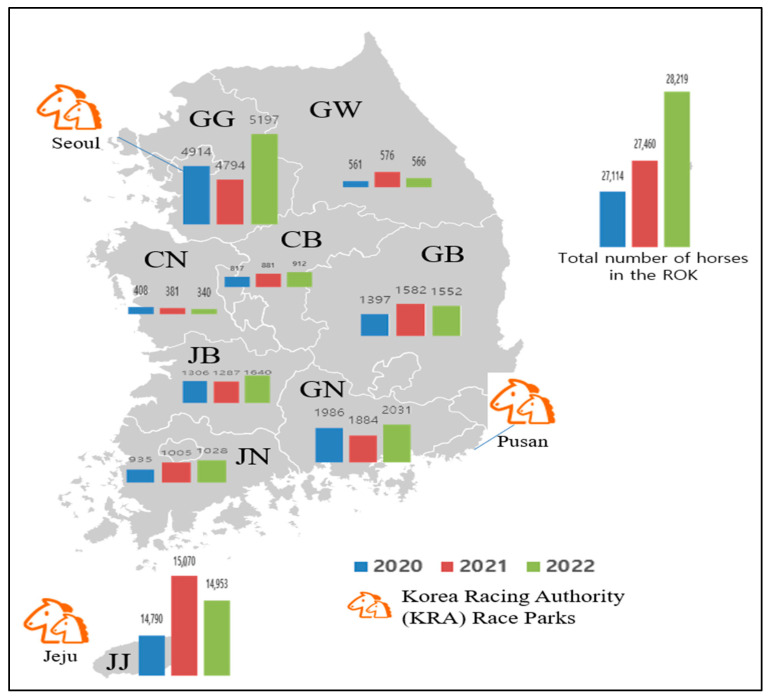
The number of horses raised in each province of the Republic of Korea (ROK) from 2020 to 2022. Gangwon, GW; Gyeonggi, GG; Chungbuk, CB; Chungnam, CN; Gyeongbuk, GB; Gyeongnam, GN; Jeonbuk, JB; Jeonnam, JN; and Jeju, JJ.

**Figure 2 viruses-15-02135-f002:**
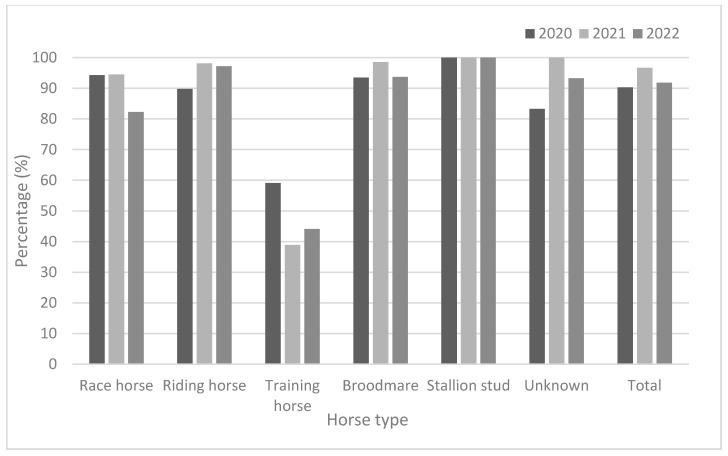
Comparison of seroprevalence according to horse type from 2020 to 2022.

**Figure 3 viruses-15-02135-f003:**
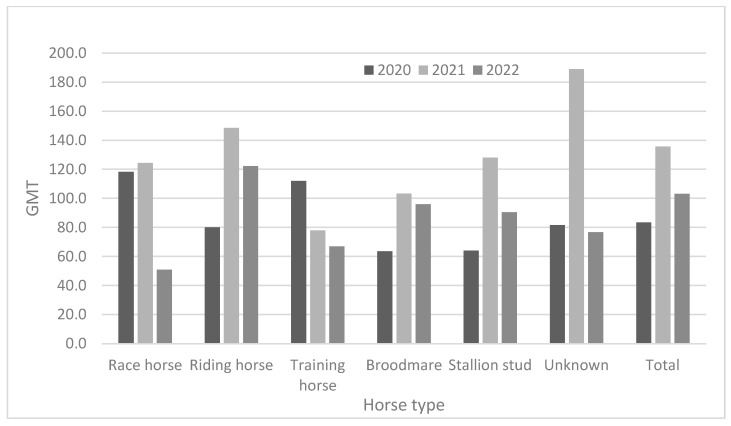
Comparison of antibody titer according to horse type from 2020 to 2022.

**Figure 4 viruses-15-02135-f004:**
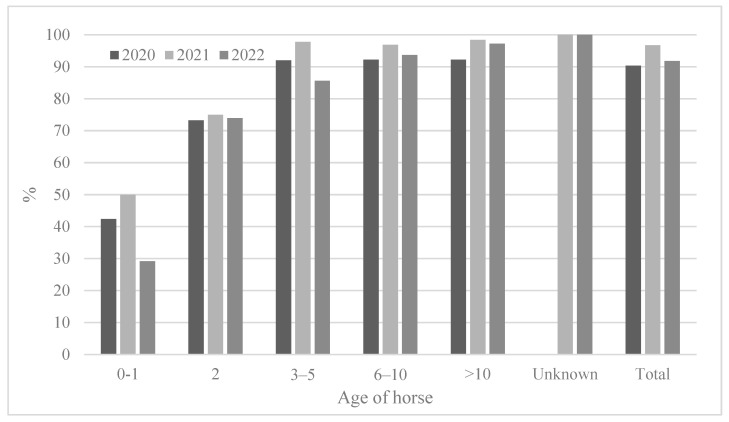
Comparison of seroprevalence according to horse age from 2020 to 2022.

**Figure 5 viruses-15-02135-f005:**
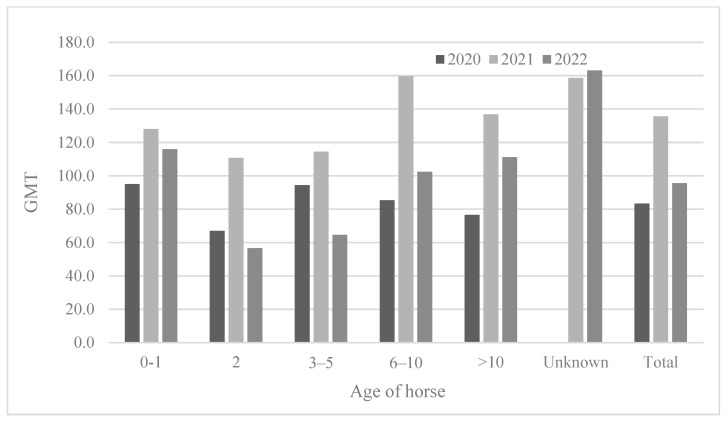
Comparison of antibody titer according to horse age from 2020 to 2022.

**Table 1 viruses-15-02135-t001:** Summary of the collected specimens from horses in the Republic of Korea (ROK) from 2020 to 2022.

Year	Number of Collected Samples		
	Blood	Nasal Swab	Lung
2020	1323	509	157
2021	1312	660	266
2022	1346	636	50
Total	3981	1805	473

**Table 2 viruses-15-02135-t002:** The seroprevalence of equine influenza virus (EIV) H3N8 (A/equine/South Africa/4/03) from 2020 to 2022.

Year	Number of Tested	Number of Positive Horse	Positive Rate (%)	GMT
2020	1323	1195	90.3	83.4
2021	1312	1269	96.7	135.7
2022	1346	1235	91.8	95.6
Total	3981	3699	92.9	103.1

**Table 3 viruses-15-02135-t003:** Antibody responses in hemagglutination inhibition (HI) test according to horse type from 2020 to 2022.

Type	2020				2021				2022			
	No. Tested	No. of Positive Horse	Positive Rate (%)	GMT	No. Tested	No. of Positive Horse	Positive Rate (%)	GMT	No. Tested	No. of Positive Horse	Positive Rate (%)	GMT
Race horse	262	247	94.3	118.3	360	351	97.5	124.3	305	251	82.3	50.9
Riding horse	725	651	89.8	80.1	685	672	98.1	148.5	722	702	97.2	122.2
Training horse	44	26	59.1	112.0	18	7	38.9	78.0	34	15	44.1	67.0
Broodmare	261	244	93.5	63.6	210	207	98.6	103.3	238	223	93.7	95.9
Stallion stud	7	7	100	64.0	7	7	100	128.0	2	2	100	90.5
Unknown	24	20	83.3	81.6	32	32	100	189.0	45	42	93.3	76.7
Total	1323	1195	90.3	83.4	1312	1269	96.7	135.7	1346	1235	91.8	103.1

**Table 4 viruses-15-02135-t004:** Antibody responses in different age groups from 2020 to 2022.

Age	2020				2021				2022			
	No. Tested	No. of Positive Horse	Positive Rate (%)	GMT	No. Tested	No. of Positive Horse	Positive Rate (%)	GMT	No. Tested	No. of Positive Horse	Positive Rate (%)	GMT
0–1	33	14	42.4	95.1	16	8	50.0	128.0	24	7	29.2	115.9
2	41	30	73.2	67.0	32	24	75.0	110.8	23	17	73.9	56.6
3–5	339	312	92.0	94.4	369	361	97.8	114.5	355	304	85.6	64.6
6–10	371	342	92.2	85.3	352	341	96.9	159.7	335	314	93.7	102.4
>10	539	497	92.2	76.6	501	493	98.4	136.9	569	553	97.2	111.2
Unknown	0	0	0	0	42	42	100	158.6	40	40	100	163.1
Total	1323	1195	90.3	83.4	1312	1269	96.7	135.7	1346	1235	91.8	95.6

**Table 5 viruses-15-02135-t005:** Antibody responses according to province from 2020 to 2022.

Province *	2020				2021				2022			
	No. Tested	No. of Positive Horse	Positive Rate (%)	GMT	No. Tested	No. of Positive Horse	Positive Rate (%)	GMT	No. Tested	No. of Positive Horse	Positive Rate (%)	GMT
Gangwon (GW)	50	48	96.0	66.8	47	47	100	171.9	57	57	100	128
Gyeonggi (GG)	462	433	93.7	84.7	460	447	97.2	111.8	477	429	89.9	83.1
Chungbuk (CB)	61	50	82.0	88.0	28	27	96.4	134.7	18	18	100	128
Chungnam (CN)	33	21	63.6	31.0	36	35	97.2	227.3	46	45	97.8	166.3
Gyeongbuk (GB)	42	40	95.2	102.2	55	55	100	150.8	52	48	92.3	81.8
Gyeongnam (GN)	86	78	90.7	95.5	148	147	99.3	189.3	153	142	92.8	78.6
Jeonbuk (JB)	72	60	83.3	80.6	50	50	100	191.3	61	57	93.4	71.4
Jeonnam (JN)	29	28	96.6	99.9	48	48	100	209.1	44	43	97.7	176.7
Jeju Island (JJ) **	488	437	89.5	83.4	440	413	93.9	124.6	438	396	90.4	105.4
Total	1323	1195	90.3	83.4	1312	1269	96.7	135.7	1346	1235	91.8	95.6

* Nine provinces in the ROK. ** Jeju Island is a special autonomous island.

## Data Availability

The data presented in this study are available upon request from the corresponding author. The data are not publicly available due to patent applications in the ROK.

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
