# Peer review of "Assessment of Equine Influenza Virus Status in the Republic of Korea from 2020 to 2022"

_viruses, 2023, doi:10.3390/v15102135_

Round 1

Reviewer 1 Report

The manuscript presents important information regarding the effectiveness of the vaccination activities in ROK. The authors may wish to considers some of the comments below in order to address some aspects that may improve the strength of the paper.

Introduction:

Line 86-86 is not clear whether the material stated is from this study or from another study of another study as there is no citation against the assertions.

The introduction needs some amending and realigning. A brief look at the subject matters does not lead the reader to subject of the paper. Please take a look at the summary of the content of the introduction. 

Paragraph 1-4: Refers to the virus clinical presentation and epidemiology

Paragraph 5: Vaccination against EI

Paragraph 6-7: Equine population and distribution

Paragraph 8: Surveillance of EI

Paragraph 9: History of vaccination in ROK and the vaccination schedule

Paragraph 10: Isolation of H3N8 in ROK and the genetic characterization

paragraph 11: Importance of antigenic changes to the virus and importance of surveillance to control programmes

Last paragraph: "In this study, we evaluated antibody levels against H3N8 and monitored the viral RNA of EIV from 2020 to 2022. This study discusses the current status of the ROK after EI vaccination".

The aspect of the "evaluation of antibody levels" appears to represent the content of the paper. However, the aspect of monitoring may need to be re-focused in a retrospective study.  Unless the authors can elaborate the aspect of monitoring of viral RNA (as monitoring has a element of active connotation to it).

Methods/Results

The figures and the tables can be summarized because the populations are presented more than once but showing the same information throughout (Table 1 & 3)

The information in the tables appears to be similar to the information on the graphs and therefore one of the two can be dropped.

Line 135: The phrase "The PCR products were sequenced to 135 confirm EIV amplification" appears to be contract Line 170: All nasal swabs and lung tissues tested negative for EIV RNA using RT- PCR.  

Line169: Please revisit the citation [9] which refers to Kim et al and not Lee et al cited. The only Lee et al is found in citation 16 (Choi, E.J.; Lee, C.H.; Song, J.Y.; Shin, Y.K.; Moon, J.S.; Choi, Y.G.; Kim, H.P)

Discussion
The discussion can be improved by aligning to the discussion at hand. As it is, it discusses so much about everything else that has no bearing on the investigation. The main discussion is the systematic and regular discussion, and this is what should guide the discussion. For instance, paragraph 2 (Line 136-237) is talking about the countries that are free and those that have epizootics. However, the paragraph is expanded to include 7 citations.

Line 267-288: Similarly, this paragraph begins by mention the non-detection of Viral RNA, but continues to discuss many other aspects of seropositivity.

Line 297 and 335: This statement is exactly. “However, 2-year-olds or younger horses may need more attention in vaccination against EIV during the vaccination regime.” most of the repetitions may be removed to improve conciseness.

Conclusions

The conclusion could be improved because in combines everything else in the manuscript. this is a suggested conclusion:

This study to evaluate the status of EIV using antibody levels against the EI vaccine confirms that EI vaccination shows showed high seropositivity rates and antibody titers. Therefore, continuous vaccination against EIV should be practiced.

Line 337: The "continuous surveillance of EIV should be practiced to monitor the introduction of this disease and EIV vaccines should be used with adequate strains" should not be part of the conclusion as this is not what the authors set out to do.

Reviewer 2 Report

This is an interesting review of EI that includes commendable surveillance work and testing in RO Korea

In this respect it does neither comprehensively sadly

It would be better to focus on the situation and application to testing performed in Korea rather than a semi review of EI 

And actively reduce introduction and discussion to more relevant aspects regarding the study results.

With a specific view to Concentrating on aspects relevant to the study

including reasons for possible lower seroprevalence in younger horses and reasons for vaccine failure/reduction in efficacy

In short the authors need to contextualise their findings in the light of current information about EIV and particularly vaccine efficacy

I have made some suggestions however please review the focus of the discussion to concentrate on aspects relevant to results

Some discussion areas need better grammar 
